

# Risk factors for hypocalcemia after total thyroidectomy: a narrative review

Bohan Cao[1] and Guangzhe Wu[2]

[1] General Surgery, China Medical University, Shenyang, Liaoning, China
[2] General Practice, General Hospital of Northern Theater Command, Shenyang, Liaoning, China

## ABSTRACT

Hypocalcemia is a frequent complication after total thyroidectomy and seriously affects patients' postoperative quality of life and long-term prognosis. This article reviews the relationship between postoperative hypocalcemia and its suspected risk factors, including age, sex, serum magnesium, vitamin D, high-risk pathological subtype, parathyroid injury, and parathyroid hormone levels, and assesses the ability of preoperative and postoperative parathyroid hormone levels and changes therein at various time points to predict postoperative hypocalcemia. It also discusses the protection of the parathyroid glands in situ by tracer techniques during total thyroidectomy. The various studies that have concluded that parathyroid injury is the most important indicator of postoperative hypocalcemia among these risk factors are reviewed. It is important for general surgeons to know how to avoid intraoperative parathyroid injury, which will contribute to the prevention of hypocalcemia.

## INTRODUCTION

Thyroid cancer is the most prevalent malignant tumor in the neck, with a higher incidence in women (*Pizzato et al., 2022*). The incidence of thyroid cancer is rising, making it one of the most common malignancies in China (*Wu et al., 2022*). Surgery remains the preferred treatment option (*Giuffrida et al., 2019*), and total/near-total thyroidectomy (TT) should be performed in the following cases: when the tumor diameter is greater than four cm; in cases of bilateral multifocal carcinoma; in cases with a high-risk pathological subtype, such as diffuse sclerotic, high-cell, and columnar papillary thyroid carcinoma, extensive invasive follicular thyroid carcinoma, and hypo-differentiated thyroid carcinoma; in the presence of cervical lymph node metastases or distant metastases; and cases with extraglandular invasion (*e.g.*, into the nerves, trachea, or esophagus) (*Mitchell et al., 2016*). Similar to other surgical operations, patients who undergo TT might face postoperative complications such as hypocalcemia, bleeding, local hematoma, injury to the recurrent or superior laryngeal nerve, and lymphatic leakage. Among these, hypocalcemia is the most frequent complication following TT, with an occurrence rate ranging between 25% and 40% (*Rocke et al., 2020*; *Casey & Hopkins, 2023*). Hypocalcemia is defined as a serum calcium concentration below the normal range of 2.10–2.60 mmol/L (*Pepe et al., 2020*).

Corresponding author
Guangzhe Wu, wgz406@163.com

Subclinical hypocalcemia refers to a condition in which serum calcium levels fall below the normal reference range without obvious clinical symptoms. It is usually found in routine tests after surgery and may suggest mild or early parathyroid problems, which could associated with long-term risks (*Orloff et al., 2018*). According to the American Thyroid Association's 2016 guidelines, hypocalcemia after TT is categorized as transient when it resolves within 6 months of surgery, whereas it is considered permanent if it persists beyond 6 months postoperatively (*Shoback et al., 2016*). Hypocalcemia may be an asymptomatic laboratory finding or a very worrying long-term adverse outcome after TT (*Van Slycke et al., 2021*). The most frequent clinical manifestations of symptomatic hypocalcemia are muscle twitching or spasms and anxiety. However, clinical manifestations sometimes take the form of severe or even life-threatening metabolic disorders, as well as laryngospasm, seizures, cardiac arrhythmias, and heart failure (*Bilezikian, 2020*). Although hypocalcemia after TT is transient in most cases, approximately 5%–10% of patients develop permanent hypocalcemia, which has profound effects on postoperative quality of life and the long-term prognosis and requires lifelong calcium and vitamin D supplementation (*Rocke et al., 2020*). Even transient hypocalcemia can have adverse effects (*Doubleday et al., 2021*). Therefore, how to predict the risk of postoperative hypocalcemia has become an urgent problem. As shown in Table 1, an increasing number of potential risk factors for postoperative hypocalcemia, including age, sex, serum magnesium, serum vitamin D, parathyroid injury (*e.g.*, contusion, impaired blood supply, inadvertent removal of the parathyroid glands), and the parathyroid hormone (PTH) level, have been investigated in recent years (*Qin et al., 2020*).

The aim of this review is to identify hypocalcemia after TT early and even prevent it by analyzing the relationship between hypocalcemia and these risk factors. Clinicians and researchers in the field of surgery and in the field of endocrinology are provided with references and support.

## SURVEY METHODOLOGY

This study involved a search of two databases, PubMed and China National Knowledge Infrastructure, from January 2010 to December 2024. Using the following key terms and phrases: (1) ''total thyroidectomy'' as either subject headings or free terms, combined with ''hypocalcemia/hypoparathyroidism'' or their free terms; (2) ''total thyroidectomy'' or its free terms combined with ''hypocalcemia/hypoparathyroidism'' or their equivalents; (3) Keywords and subject headings related to ''total thyroidectomy'', ''hypocalcemia/hypoparathyroidism'' and/or ''risk factors''. An initial screening of article titles was conducted, followed by a more detailed secondary review. The inclusion criteria for the articles reviewed were as follows: (1) Original studies (prospective/retrospective cohorts, case-control) reporting risk factors for post-TT hypocalcemia, sample size ≥50 patients; (2) articles published in English or Chinese regardless of study design; (3) clear definitions of hypocalcemia (*e.g.*, serum calcium < 2.10 mmol/L or symptomatic). (4) articles deemed most relevant for this review.

The Exclusion Criteria: (1) Case reports, or non-human studies; (2) studies lacking multivariate analysis for risk factors; (3) duplicate publications or overlapping datasets.

**Table 1** Summary of the included studies of risk factors for hypocalcemia after total thyroidectomy in patients with thyroid cancer.

| Risk factors | Cases | Design | Conclusions | References |
|---|---|---|---|---|
| Age | 200 | Prospective cohort study | The risk for postoperative hypocalcemia was increased 20-fold for patients older than 50 years. | *Tolone et al. (2013)* |
| | 68 | Retrospective study | Children with thyroid cancer are at high risk for postoperative hypocalcemia after total thyroidectomy. | *Zobel et al. (2020)* |
| Sex | 2,108 | Retrospective study | Female gender is a strong risk factor that influence early hypocalcemia development. | *Del Rio et al. (2019)* |
| | 734 | Retrospective study | Young female patients undergoing neck dissection are at higher risk of developing temporary hypoparathyroidism. | *Privitera et al. (2023)* |
| Magnesium | 312 | Prospective cohort study | Serum magnesium below 1.9 mg/dL had 2.7 times higher odds of developing transient hypocalcemia post-TT. | *Karunakaran et al. (2020)* |
| Vitamin D | 181 | Retrospective cohort study | A correlation between transient postoperative hypocalcemia and 25-hydroxyvitaminD levels. | *Saibene et al. (2022)* |
| | 100 | Retrospective cohort study | Preoperative serum Vit. D levels did not affect postoperative serum calcium levels. | *Layegh et al. (2024)* |
| Pathological type of thyroid cancer | 453 | Retrospective randomized controlled study | TT + iCCND is associated with a significantly increased risk of transient hypoparathyroidism and TT + bCCND is associated with a significantly increased risk of transient and permanent hypoparathyroidism. | *Rosati et al. (2022)* |
| Parathyroid injury | 657 | Meta-analysis | There is a linear relationship between parathyroid glands preserved in situ and the prevalence of all hypoparathyroidism syndromes. | *Sitges-Serra (2021)* |
| | 244 | Retrospective study | Parathyroid autotransplantation does not influence the rate of postoperative hypocalcemia and/or hypoparathyroidism. | *Tartaglia et al. (2016)* |
| | 1,870 | Meta-analysis | The application of CNs in total or near-total thyroidectomy combined with CLND for thyroid cancer can better dissect the central lymph nodes and protect parathyroid glands and their function. | *Zhang et al. (2023)* |

**Table 1** (*continued*)

| Risk factors | Cases | Design | Conclusions | References |
|---|---|---|---|---|
| Parathyroid protection | 1,711 | Meta-analysis | The reduced risk of postoperative hypoparathyroidism and hypocalcemia reflected NIRAF preservation value. | *Safia et al. (2024)* |
| | 90 | Prospective observational study | ICG angiography of the parathyroid gland is a safe, reliable predictor for postoperative transient hypocalcemia. | *Abdelrahim, Amer & Mikhael Nageeb (2022)* |
| PTH | 521 | Retrospective study | POD1 PTH levels $\geq$ 15 pg/ml along with calcium $\geq$ 2.0 mmol/l are associated with low risk of symptomatic hypocalcemia. | *Riordan et al. (2022)* |
| | 1,636 | Retrospective cohort study | Approximately one-quarter of all patients with low PTH levels immediately after surgery developed permanent hypoparathyroidism. | *Annebäck et al. (2024)* |
| | 87 | Retrospective cohort study | Postoperative 4 h PTH to preoperative PTH ratio with a cutoff point around 0.385 is an excellent indicator for identifying patients at risk for postoperative hypocalcemia. | *Daskalaki et al. (2022)* |

**Notes.**
bCCND, bilateral central compartment neck dissection; CLND, complete lymph node dissection; CNs, carbon nanoparticles; iCCND, ipsilateral central compartment neck dissection; ICG, indocyanine green; NIRAF, near-infrared autofluorescence; POD1, postoperative day 1; PTH, parathyroid hormone; TT, total thyroidectomy.

Finally, the full texts were obtained for further evaluation. Additionally, references from relevant articles were reviewed to identify other eligible studies.

## LITERATURE REVIEW

### Patient age

The influence of age on the risk of hypocalcemia after TT remains controversial. However, most studies have concluded that the incidence of hypocalcemia is significantly higher in the elderly, likely because of the physiological characteristics of aging, cumulative damage to regulation of the gastrointestinal, metabolic, immune, and endocrine systems. This damage leads to a decrease in intestinal absorption of calcium, as well as metabolic abnormalities, including decreases in the activity of renal 1-hydroxylase and in accumulation of 7-dehydrocholesterol in the skin. Therefore, advanced age may be the main risk factor for the occurrence of postoperative hypocalcemia (*Tolone et al., 2013*; *Eismontas et al., 2018*). However, recent studies have indicated that young age may also be a risk factor, given the observation that extraglandular invasion and central lymph node metastasis are more prevalent in younger patients with thyroid cancer, including children. Furthermore, these young patients are more likely to undergo central lymph node dissection (CLND), which significantly increases the risk of hypocalcemia and hypoparathyroidism (HPT) (*Zobel et al., 2020*; *Spinelli et al., 2022*). In conclusion, the relationship between age and the risk of

postoperative hypocalcemia is non-linear, in that postoperative hypocalcemia tends to be transient in young people and is more likely to be permanent in the elderly.

## Patient sex

Although the results of the available studies are not entirely consistent, hypocalcemia tends to be more likely to occur in women after thyroidectomy, so female sex may be an independent risk factor. This is because the female parathyroid glands (PTGs) are usually small, have a fine vascular lumen, and are likely to be located within the thyroid capsule, which increases the risk of intraoperative injury. Moreover, there may be sex-related differences in the effects of steroid hormones on secretion of PTH and genetic variations in cell signaling pathways, resulting in greater susceptibility to hypocalcemia in female patients (*Villarroya-Marquina et al., 2020*).

Studies in the literature examining gender as a risk factor indicates that postoperative hypocalcemia following TT is more prevalent in women than in men. *Docimo et al. (2017)* reported a significant correlation between being female and reduced postoperative serum calcium levels, though its relevance to symptomatic hypocalcemia remains unclear. *Sands et al. (2011)* suggested that female sex could be a potential risk factor for transient postoperative hypocalcemia, irrespective of menopausal status; however, additional research is required to assess the clinical significance of this observation. Similarly, *Del Rio et al. (2019)* highlighted female sex as a major risk factor for early postoperative hypocalcemia. A more radical view put forward is that women aged <45 years are more prone to hypocalcemia and even HPT because of the effect of estrogens on secretion of PTH in premenopausal women (*Villarroya-Marquina et al., 2020*; *Privitera et al., 2023*).

## Serum magnesium

The serum magnesium concentration affects the serum calcium level mainly by competing with calcium ions for binding sites, and hypomagnesemia leads to HPT by inhibiting both the secretion of PTH from the PTGs as well as the responsiveness of target organs (*Mahmoud et al., 2016*). One retrospective study found that hypomagnesemia was relatively common after TT, with an incidence of approximately 70% (*Cherian et al., 2016*). The elevated incidence was linked to the release of antidiuretic hormone triggered by surgical stress, leading to sodium retention, hemodilution (regardless of infusion volume), and a temporary reduction in serum magnesium levels. This indicates that hemodilution is a key factor in the development of postoperative hypomagnesemia. Hypomagnesemia shows a strong correlation with the symptoms and manifestations of postoperative hypocalcemia, and even serum magnesium levels at the lower end of the normal range can heighten the risk of hypocalcemia. Another study concluded that patients with a postoperative serum magnesium level <1.9 mg/dL had a 2.7-fold increase in likelihood of transient hypocalcemia and noted that hypomagnesemia in the first 48 h postoperatively was associated with refractory hypocalcemia and permanent HPT (*Karunakaran et al., 2020*). However, the study was small and further research is needed.

## Vitamin D

Vitamin D, a fat-soluble vitamin synthesized from cholesterol, undergoes hydroxylation in the liver to produce 25-hydroxyvitamin D [25(OH)D]. Subsequently, it is converted into its active form, 1,25-dihydroxyvitamin D [1,25(OH)2D], in the kidney through a process regulated by PTH (*Dai et al., 2018*). Vitamin D enhances calcium absorption in the intestine and can be beneficial for patients with hypocalcemia, provided there is no underlying malabsorption. Additionally, it promotes bone resorption while decreasing renal excretion of both calcium and phosphate. When vitamin D levels are deficient, PTH synthesis is upregulated, which further facilitates calcium absorption (*Dugani et al., 2023*).

The role of vitamin D in postoperative hypocalcemia remains a topic of debate. Certain researchers have proposed a link between low preoperative serum vitamin D levels and the occurrence of postoperative hypocalcemia (*Bove et al., 2020*) identified that preoperative vitamin D deficiency (VDD), defined as a 25(OH)D level below 25 ng/mL, serves as an independent predictor of transient hypocalcemia after TT. A meta-analysis states that VDD is associated with an increased risk of transient hypoparathyroidism after thyroidectomy, while only severe VDD is linked to a higher likelihood of permanent hypoparathyroidism (*Vaitsi et al., 2021*). Moreover, *Saibene et al. (2022)* found that VDD was a significant predictor of transient postoperative hypocalcemia (odds ratio 0.343), which was confirmed by *Choi et al. (2021)*. *Vibhatavata et al. (2020)* found that a preoperative vitamin D level < 19.6 ng/mL predicted symptomatic postoperative hypocalcemia with a sensitivity of up to 82%. Paradoxically, there are other researchers who have taken the opposite view. *Soares, Tagliarini & Mazeto (2021)*, *Sitges-Serra (2021)* concluded that the preoperative vitamin D level is not predictive of postoperative hypocalcemia. *Singh et al. (2021)* agreed with this view but noted that the variable findings regarding this risk factor may reflect the lack of multicenter randomized controlled trials. There is also recent evidence indicating that preoperative serum vitamin D status does not affect the postoperative calcium level (*Wang et al., 2017b*; *Layegh et al., 2024*). In patients with Graves' disease (GD), VDD is common due to increased bone turnover and significantly elevates the risk of postoperative hypocalcemia. A prospective study reported symptomatic hypocalcemia in 65% of patients with 25(OH)D < 20 ng/mL *versus* 28% in those with sufficient levels (*Al-Khatib et al., 2015*).

Despite these controversies, it is likely that perioperative prophylactic calcium or vitamin D supplementation can prevent transient and symptomatic hypocalcemia and potentially shorten the hospital stay (*García Pascual et al., 2023*; *Jan et al., 2024*). It has recently been found that the calcium-raising effect of 1,25(OH)2D is more persistent than that of PTH because of its significantly longer half-life and the fact that the 1α-hydroxylase activity of 25(OH)D is maintained to some extent in the face of reduced PTH. Therefore, compared with a reduced PTH level, a decreased 1,25(OH)2D level is a relatively greater risk factor for postoperative hypocalcemia after TT (*Yamashita et al., 2024*).

## Pathological subtype

High-risk thyroid cancer concomitant with Hashimoto's thyroiditis leads to adhesion and unclear boundaries between the thyroid gland and the surrounding tissues. In this

situation, the malignant tumor can directly infiltrate into the thyroid capsule and even invade the esophagus and trachea, such that removal of the PTGs and surrounding tissues is necessary to achieve adequate safety margins (*Kuroya et al., 2020*). Patients with thyroid cancer undergoing CLND are at an increased risk of developing hypocalcemia, primarily due to tumor invasion that compromises the blood supply or necessitates the unavoidable removal of the PTGs.

According to the American guidelines for diagnosing and managing differentiated thyroid cancer, TT involves CLND on at least one side of the thyroid gland. The dissection boundaries are defined as extending superiorly to the hyoid bone, inferiorly to the level of the innominate artery, laterally to the common carotid artery, anteriorly to the superficial layer of the deep cervical fascia, and posteriorly to the deeper layer of the deep cervical fascia. Lateral cervical lymph node dissection is indicated when metastatic involvement is detected in the lateral cervical region.

Routine dissection of the lateral cervical region includes zones II to V, extending superiorly to the digastric muscle, inferiorly to the superior edge of the clavicle, medially to the carotid sheath, and laterally to the anterior border of the trapezius muscle (*Haugen et al., 2016*). There is some literature suggesting that the incidence of postoperative hypocalcemia is lower when TT is performed without CLND (*Harris et al., 2016*; *Lale et al., 2019*). In 2017, a meta-analysis found that CLND undoubtedly increases the incidence of both transient and permanent hypocalcemia (*Zhao et al., 2017*). Furthermore, a study in India from 2022 found that ipsilateral central neck dissection increased the risk of transient HPT while bilateral central neck dissection increased the risk of both permanent HPT and hypocalcemia (*Rosati et al., 2022*).

GD carries a higher risk of postoperative hypocalcemia (*Hallgrimsson et al., 2012*; *Liang et al., 2023*); *Liu et al. (2011)* demonstrated that GD patients undergoing TT had a 1.5-2-fold higher incidence of transient hypocalcemia compared to those with multinodular goiter or thyroid cancer. This elevated risk is attributed to the hypervascularity and friability of the thyroid tissue, making parathyroid preservation more challenging (*Ross et al., 2016*). Optimal perioperative management is crucial for minimizing the risk of postoperative hypocalcemia in patients with GD (*Chuang et al., 2022*; *Reinke et al., 2023*; *Elkhoury et al., 2023*).

## Parathyroid glands integrity and parathyroid hormone
### Parathyroid glands injury
Hypocalcemia may develop in patients with compromised parathyroid gland function. The primary intraoperative causes of HPT after TT include contusions, devascularization, and accidental removal of the parathyroid glands (IPE) (*Isaksson et al., 2019*). Surgical procedures that include crushing, burning, clamping, and suturing can result in contusions within the PTGs, which can affect their functioning. Furthermore, the parathyroid artery is a single vessel that is delicate and fragile. Intraoperative manipulation (*e.g.*, traction or direct injury) can cause vasospasm and local thrombosis, resulting in disruption of the blood supply to the PTGs. IPE is relatively common in thyroid surgery and has several causes, including variations in the anatomical location of the PTGs, inexperience on the

part of the surgeon, and local inflammation. The direct relationship between IPE and hypocalcemia/HPT remains controversial. Some retrospective studies have concluded that IPE does not lead to an increased incidence of hypocalcemia if more than three PTGs are preserved *in situ* and that the risk of permanent HPT may be significantly increased if fewer than two PTGs are preserved (*Du et al., 2017*; *Karadeniz & Akcay, 2019*). A more recent meta-analysis published in 2021 concluded that IPE may increase the prevalence of permanent HPT and hypocalcemia by 5%–8% (*Sitges-Serra, 2021*). Parathyroid autotransplantation is used to preserve parathyroid function after parathyroid injury or treat problems with blood supply, and usually involves transplantation into the brachialis muscle of the forearm with routine frozen sections. However, research has shown that parathyroid autotransplantation does not prevent postoperative hypocalcemia (*Tartaglia et al., 2016*).

### Preservation of the parathyroid glands

The most effective method to maintain normal parathyroid function postoperatively is to retain the PTGs in their natural position, avoiding any intraoperative injury. The inferior parathyroid glands (IPTG) are challenging to identify during cervical lymph node dissection, particularly when dealing with central neck dissection, because of their irregular location and the complex anatomical features of the surrounding tissues (*Cui et al., 2016*; *Wang et al., 2017a*; *Hou et al., 2020*; *Rao et al., 2023*). Although use of a thymus–blood vessel–IPTG layer (also known as meticulous capsular dissection) to increase *in situ* preservation of the IPTG is now a viable option (*Wang et al., 2017a*; *Wang et al., 2022*; *Hou et al., 2020*; *Rao et al., 2023*), distinguishing the IPTG remains challenging for surgeons in some patients. Therefore, identification of the IPTG at the time of lymph node dissection is essential for reduction of the risk of postoperative hypocalcemia. Intraoperative parathyroid venous congestion is also an important but sometimes overlooked cause of parathyroid dysfunction and subsequent hypocalcemia following TT. Venous congestion of the PTGs, even in the presence of preserved arterial inflow, can lead to glandular edema, impaired microcirculation, and ultimately hypoparathyroidism if not promptly recognized and managed. Effective intraoperative management strategies include meticulous identification and preservation of parathyroid vasculature, minimizing manipulation, and careful dissection close to the thyroid capsule to avoid injury to the delicate venous outflow pathways. When venous congestion is identified, typically manifesting as dark discoloration and swelling of the gland, timely intervention is crucial. In cases where compromised vascular integrity or iatrogenic injury jeopardizes the *in situ* perfusion of PTGs, autologous parathyroid transplantation becomes clinically indicated. Historically, surgical assessment of PTG viability relied on subjective visual assessment of glandular coloration and tissue turgor (*Lang et al., 2016*). However, the advent of intraoperative angiography has revolutionized parathyroid preservation strategies, with contemporary thyroid surgeons increasingly employing this technology for real-time visualization of microvascular perfusion and objective evaluation of parathyroid viability during endocrine neck procedures (*Park et al., 2016*; *Lang et al., 2017*). One imaging technique is indocyanine green fluorescence (ICGF). This method is mainly used to identify the PTGs *in situ* using

a safe and rapidly metabolized fluorescent dye to assess the viability of the PTGs by blood perfusion and to guide autotransplantation (*Barbieri et al., 2021*; *Yin et al., 2022*; *Moreno Llorente et al., 2022*; *Abdelrahim, Amer & Mikhael Nageeb, 2022*). The ICGF strategy has been demonstrated to exhibit optimal efficiency in augmenting the rate of PTGs auto-transplantation when contrasted with other intraoperative visualization of PTGs strategies (*Lu et al., 2024*). Recent studies have shown that an indocyanine green-macroaggregated albumin-hyaluronic acid mixture (ICG-MAA-HA), also known as LuminoMark™, is effective in locating recurrence of thyroid cancer and has been shown to reduce the adverse effects of surgery (*Kim et al., 2024*). Another is near-infrared autofluorescence (NIRAF) imaging, which detects parathyroid autofluorescence with 97%–99% consistency because autofluorescence is stronger in the PTGs than in surrounding tissues and has been used intraoperatively to accurately identify the PTGs in real time (*Lu et al., 2022*; *Barbieri et al., 2023b*; *Barbieri et al., 2023a*; *Safia et al., 2024*). Other tracer technologies can also be used for accurate identification of the PTGs to protect their function (*Rao et al., 2023*). Carbon nanoparticles (CNs) are frequently employed as tracers in China, capable of entering lymphatic vessels but not capillaries. As a result, when injected into thyroid tissue, these nanoparticles selectively stain the draining lymph nodes black, thereby aiding in the lymph node dissection process (*Yu et al., 2018*). They do not enter the circulation, so the PTGs and peripheral blood vessels are not stained, which helps the operator to identify the PTGs (*Zhang et al., 2023*). A recent meta-analysis showed that CNs reduce the frequency of postoperative hypocalcemia and HPT by reducing the risk of IPE (*Wang et al., 2023*). However, some studies have found that they do not significantly reduce the incidence of permanent HPT or hypocalcemia (*Xue et al., 2018*). A study conducted at Xi'an Jiaotong University (China) demonstrated that the use of CNs assisted in identifying the PTGs and regional lymph nodes. However, it did not prevent a postoperative decrease in intact parathyroid hormone levels nor did it lower the incidence of hypocalcemia or HPT after TT (*Liu et al., 2020*). The authors of that report concluded that CNs did not improve the clinical outcome and that their value should not be overstated by experienced thyroid surgeons working in tertiary hospitals. ICGF, NIRAF, CNs are all visualization strategies that offer significant advantages over subjective visual assessment during thyroid surgery. A meta-analysis has demonstrated that each modality provides distinct benefits. ICGF has been shown to be the most effective technique for assessing PTGs viability intraoperatively. NIRAF is particularly effective in reducing the incidence of postoperative hypocalcemia, enhancing PTGs identification, and decreasing the rates of IPE and autologous parathyroid transplantation. Meanwhile, CNs have demonstrated a beneficial role in lowering the incidence of postoperative hypoparathyroidism (*Lu et al., 2024*). The above studies suggest that imaging technique-guided thyroidectomy reduces the incidence of postoperative hypocalcemia (*Demarchi et al., 2021*), but more high-quality studies are needed to confirm this effect.

### Parathyroid hormone

PTH is a peptide hormone produced and released by the PTGs, primarily affecting target organs such as the bones and kidneys, and plays a key role in regulating calcium

and phosphate metabolism. Serum calcium levels in the human body are primarily regulated by PTH and vitamin D. Since PTH serves as the primary biochemical marker for assessing serum calcium, it has been extensively investigated as a potential risk factor for hypocalcemia after TT (*Mazotas et al., 2018*; *Mo et al., 2020*; *Privitera et al., 2021*). The method most commonly used to detect postoperative hypocalcemia and HPT is still the second-generation intact PTH assay (*Orloff et al., 2018*). This assay detects both PTH(1-84) and larger N-truncated fragments, primarily PTH(7-84). The action of PTH(7-84) opposes that of PTH(1-84), which suggests that the intact PTH assay may not accurately reflect parathyroid function. While the plasma half-life of intact PTH(1-84) is only a few minutes, the renal clearance of PTH fragments occurs at a slower rate (*Inaba et al., 2004*; *Bieglmayer et al., 2006*; *Eastell et al., 2014*). Furthermore, an increased serum calcium level or 1,25(OH)2D concentration stimulates release of more N-truncated PTH fragments than PTH(1-84) from the parathyroid gland. The third-generation bio-intact PTH assay focuses on PTH(1-84) (*Inaba et al., 2004*; *Eastell et al., 2014*). Patients with renal failure, hypercalcemia, and hyperparathyroidism have high intact PTH (*Bieglmayer et al., 2006*; *Eastell et al., 2014*; *Einbinder et al., 2017*). The third-generation assay can have good results in patients with primary and secondary hyperparathyroidism, but may not be superior to intact PTH for diagnosis of HPT and other conditions associated with hypocalcemia (*Clarke et al., 2016*). Therefore, in patients without renal abnormalities after TT, intact PTH may not affect the accuracy of prediction of postoperative hypocalcemia. Recent studies examining the role of postoperative intact PTH levels and their changes in predicting hypocalcemia have assessed PTH levels at various intervals, ranging from intraoperative skin closure to 48 h after surgery, and even up to 4 days postoperatively. However, there remains no agreement on the ideal timing for perioperative PTH testing or the correct predictive threshold.

### Postoperative measurement

Postoperative measurement of intact PTH is widely used to assess hypocalcemia after TT, and numerous studies have investigated the optimal time for measurement. As early as 2007, *Sywak et al. (2007)* found that a low PTH level (3–10 pg/mL) at 4 h postoperatively had good diagnostic accuracy for prediction of postoperative hypocalcemia with 90% sensitivity and 84% specificity. They also found that the diagnostic accuracy of the PTH concentration at 23 h postoperatively was not significantly different from that at 4 h after surgery. Many studies subsequently measured the PTH concentration at 4 h postoperatively. These studies were summarized in 2017 by *Mazotas & Wang (2017)*, who noted that the PTH level measured within 4 h after surgery accurately predicted development of hypocalcemia. The sensitivity, specificity, and both positive and negative predictive values were comparable between the threshold PTH level of 10 pg/mL on postoperative day 1 (POD1) and 4 h after surgery, and combining these measurements did not enhance the accuracy of predicting hypocalcemia after TT. Therefore, additional laboratory tests may not be necessary. The PTH on POD1 is the more traditional predictor of postoperative hypocalcemia. *Selberherr et al. (2015)* prospectively stratified patients undergoing TT without lymph node dissection according to the PTH value on POD1, classifying them as having "normal" parathyroid

 

function (PTH > 15 pg/mL), "disturbed" function (PTH < 10 pg/mL), or "indeterminate" function (PTH 10–15 pg/mL). They found that patients with a PTH < 10 pg/mL were at highest risk for postoperative hypocalcemia with a sensitivity of 83% and specificity of 99%. *Cayo et al. (2012)* also found that a PTH of <10 pg/mL on POD1 had a sensitivity of 86% and a negative predictive value of 90% for symptomatic hypocalcemia. *Godlewska et al. (2020)* study suggested that undetectable PTH (<6 pg/mL) on POD1 was a significant risk factor for permanent hypocalcemia after TT + CLND. *Riordan et al. (2022)* concluded that a PTH threshold of 15 pg/mL on POD1 appeared to reliably identify low-risk patients with symptomatic hypocalcemia.

### Intraoperative measurement

Intraoperative PTH monitoring has been increasingly used in recent years to predict hypocalcemia after TT. *Lang, Yih & Ng (2012)* conducted a study in which they assessed the PTH level immediately upon skin closure (approximately 5–10 min following thyroidectomy, while the patient remained under anesthesia), a process they referred to as the "rapid measurement of intact PTH at skin closure". They found this measurement to have a sensitivity of 82.4% and a specificity of 95.0% for detection of symptomatic or biochemical hypocalcemia (<1.9 mmol/L) and suggested that it was likely to be more specific and predictive of the PTH level than the level measured on the morning after surgery. A similar study was conducted by *Reddy et al. (2016)*, who monitored intact PTH while patients were still under anesthesia for 20 min after surgery and found it to have a sensitivity of 91% and a specificity of 93% for prediction of symptomatic postoperative hypocalcemia. The results of these two studies suggest that intraoperative PTH can be used to predict hypocalcemia, assess the day of discharge, and guide the decision regarding calcium supplementation after surgery. In China, immunocolloid gold chromatography test paper has been used in the intraoperative PTH assay to aid in assessment of parathyroid function and reduce the incidence of hypocalcemia. This test takes 12–15 min to detect PTH and has a short waiting time with an accuracy rate of 98.29% (*Wei et al., 2019*; *Solórzano et al., 2021*). Recent studies have shown that almost a quarter of patients with a low PTH level in the immediate postoperative period develop permanent HPT and hypocalcemia (*Annebäck et al., 2024*).

### Preoperative measurement

Currently, preoperative intact PTH measurement is mainly used for comparison with postoperative PTH levels (*Sands et al., 2011*) found that if the postoperative PTH level was decreased by 70% or more in comparison with the preoperative PTH, it had a sensitivity of 91.2% and a specificity of 98% for predicting postoperative hypocalcemia. *Kakava et al. (2020)* used a similar method to investigate risk factors for postoperative HPT and obtained positive results. A more recent study in 2022 concluded that the ratio of the postoperative PTH at 4 h to the preoperative PTH is an excellent indicator of symptomatic hypocalcemia in the early postoperative period and would be a valuable addition for early identification of patients at high risk of hypocalcemia (*Daskalaki et al., 2022*). These researchers suggest that a decrease in the ratio of the short-term postoperative PTH to the preoperative PTH

should be used as soon as possible as an indicator when assessing whether to implement postoperative calcium supplementation.

### Rationale for investigating additional risk factors

PTH levels remain a common predictor of hypocalcemia after TT, reliance solely on PTH has limitations. First, conclusive evidence regarding the ideal timing for measurements and the predictive threshold is lacking (*e.g.*, 4 h *vs.* 24 h postoperatively) (*Mazotas et al. 2018*; *Riordan et al., 2022*). Second, transient PTH suppression due to intraoperative manipulation may obscure true parathyroid function (*Lang, Yih & Ng, 2012*). Third, identifying additional risk factors (*e.g.*, preoperative VDD or GD) enables early risk stratification and prophylactic interventions, even before PTH results are available. Ultimately, a multifactorial risk model combining PTH with demographic, biochemical, and surgical factors improves predictive accuracy and guides personalized care. This approach is particularly relevant for high-risk subgroups (*Liu et al., 2011*).

## CONCLUSIONS

TT is undoubtedly a beneficial life-saving operation, but permanent hypocalcemia after TT has an adverse impact on the prognosis and becomes a lasting burden for thyroid surgeons. Age, sex, serum magnesium, vitamin D, high-risk pathological subtype, parathyroid injury, and the PTH level may be associated with hypocalcemia after TT but the evidence is still controversial and sometimes contradictory. At present, the method most commonly used to predict hypocalcemia is the PTH test. There have been many studies on the PTH test performed before and 20 min, 1, 4, 6, 8, and 23 h, 1 day, and even 4 days after surgery. However, conclusive evidence regarding the ideal timing for measurements and the predictive threshold is lacking. Future studies should focus on determining the most reliable indicators of postoperative hypocalcemia, including the best time for testing and the appropriate diagnostic threshold. Future research should prioritize cost-effective algorithms that integrate PTH with readily available clinical data to optimize outcomes across diverse healthcare settings. Considering the importance of protecting the PTGs in the prevention of postoperative hypocalcemia, it is necessary to use carbon nanoparticles, indocyanine green, near-infrared autofluorescence, or other tracer techniques to identify the PTGs intraoperatively. These techniques may also require multicenter, prospective studies with larger sample sizes to confirm their value and to establish a unified standard to prevent postoperative hypocalcemia after TT in the future.

## ACKNOWLEDGEMENTS

We thank Liwen Bianji (Edanz) for editing the English text of a draft of this manuscript.

### Funding

The authors received no funding for this work.

## Competing Interests

The authors declare there are no competing interests.

## Author Contributions

- Bohan Cao conceived and designed the experiments, performed the experiments, analyzed the data, prepared figures and/or tables, authored or reviewed drafts of the article, and approved the final draft.
- Guangzhe Wu conceived and designed the experiments, performed the experiments, analyzed the data, prepared figures and/or tables, authored or reviewed drafts of the article, and approved the final draft.

## Data Availability

This article is a literature review.

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
