# Peer review of "Risk factors for hypocalcemia after total thyroidectomy: a narrative review"

_PeerJ, doi:10.7717/peerj.19808_

## Round 0.1 · original submission · Minor Revisions

Reviewer 1 ·

Basic reporting

Sometimes it can be difficult to follow along with.

Experimental design

Overall good review of factors affecting hypocalcemia after TT. I would suggest reviewing more articles or expanding on the current review, as it is a bit short. Consider merging the PTH section and the PT gland section

Validity of the findings

Evidence only seems to show significant links to PTH levels s/p TT as it relates to hypocalcemia.
In your rationale/conclusions, you have to make the argument why figuring out another risk factor for hypocalcemia is relevant (are you arguing for more robust/earlier detection of hypocalcemia, cheaper methods than PTH measurements, etc). Since PTH measurements give us a pretty good idea of the risk of post-op hypocalcemia, you need to make a final determination/rationale on why this specific study matters.

Reviewer 2 ·

Basic reporting

-

Experimental design

-

Validity of the findings

-

Additional comments

This narrative review covers the incidence and prevalence of hypocalcemia following thyroidectomy, focusing on patient-related and perioperative risk factors that may predict post-operative hypocalcemia. The background is detailed with relevant citations, and the language is clear and succinct. This is not a meta-analysis or systematic review, but rather a limited review of two databases, including PubMed.

The review primarily includes recent literature from PubMed and a Chinese database. There is no meta-analysis, and the selection of studies from these databases is somewhat arbitrary and not well defined. Authors should elaborate on the selection criteria for the included papers, as not all published studies were chosen.

Given the inclusion of a Chinese database, authors must address potential bias and the limited generalizability of the data. Nonetheless, the review provides a sound overview of the literature on risk factors for hypocalcemia post-thyroidectomy.

·

Basic reporting

-

Experimental design

-

Validity of the findings

-

Additional comments

Good review!
Some faults:
1) Define the term “Hypocalcemia” and subclinical hypocalcemia.
2) Review more about Post-TT hypocalcemia for Graves’ disease.
3) The management of parathyroid venous congestion

---

## Round 0.2 · accepted · Accept

Thank you for addressing the reviewer comments, which has improved your manuscript. This is now ready for publication. Congratulations.

Reviewer 2 ·

Basic reporting

The responses and changes are adequate for publication

Experimental design

n/a

Validity of the findings

n/a

·

Basic reporting

fine.

Experimental design

well design

Validity of the findings

quite valuable

Additional comments

no any comment.